## [Peer Review File · Development (Cambridge, England)]

A glial cell-derived pathway directs regenerating optic nerve axons toward the optic chiasm

Beth M. Harvey, Melissa Baxter, Alexis M. Garcia and Michael Granato

DOI: 10.1242/dev.205048

Editor: Steve Wilson

Review timeline

Original submission:	20 June 2025
Editorial decision:	11 August 2025
First revision received:	20 November 2025
Editorial decision:	15 December 2025
Second revision received:	9 January 2026
Accepted:	12 January 2026

Original submission

First decision letter

MS ID#: dev.205048

MS TITLE: A glial cell derived pathway directs regenerating optic nerve axons toward the optic chiasm

AUTHORS: Beth Harvey, Melissa Baxter, Alexis M. Garcia and Michael Granato

Dear Beth,

Apologies for the delay in obtaining reviews on your manuscript. I have now received two referees' reports on the above manuscript, and have reached a decision. The referees' comments are appended below, or you can access them online: please go to: *****

As you will see, the referees express considerable interest in your work, but have some criticisms and suggestions for improvements. If you are able to revise the manuscript along the lines suggested, I will be happy to receive a revised version of the manuscript. Please also note that Development will normally permit only one round of major revision. If it would be helpful, you are welcome to contact us to discuss your revision in greater detail. Please send us a point-by-point response indicating your plans for addressing the referees' comments, and we will look over this and provide further guidance.

Please attend to all of the reviewers' comments and ensure that you clearly highlight all changes made in the revised manuscript. Please avoid using 'Tracked changes' in Word files as these are lost in PDF conversion. I should be grateful if you would also provide a point-by-point response detailing how you have dealt with the points raised by the reviewers in the 'Response to Reviewers' box. If you do not agree with any of their criticisms or suggestions please explain clearly why this is so.

Reviewer 1

Advance summary and potential significance to field

In this manuscript, Harvey et al. identify the glycosyltransferase Lh3 as essential for directional outgrowth of retinal ganglion cell (RGC) axons following optic nerve injury. The authors demonstrate that Lh3 is upregulated by neighboring tissues following nerve transection and that transgenically supplying Lh3 from oligodendrocyte lineage cells is sufficient for restoration of directional axon guidance. Finally, the authors suggest that Col18a1, which is also upregulated following transection and essential for proper nerve regeneration, is subject to modification by Lh3 to promote optic nerve regeneration. Altogether, this work identifies a pro-regenerative pathway in the CNS that is necessary to steer RGC axons along the proper path following nerve transection. This work also identifies comparable mechanisms between PNS and CNS environments, as previous work from this group found that Lh3 is also necessary for regenerating motor axons via peripheral glia-mediated pathways. Altogether, this work builds on previous studies to demonstrate recapitulation of Lh3 function across tissues to promote axon regeneration.

Comments for the author

Major comments

- The authors discuss bypassing the requirement for Lh3 during development by using a validated heat-shock promoter driving Lh3 beginning early in development, as described in methods. This note suggests that perhaps there is a developmental requirement for Lh3 by RGC axons that is not yet described, at least not in the cited papers. I have concerns that this might confound the take-home messages of this study.
 - What is the optic nerve phenotype of Lh3 mutants without hsp:Lh3 expression? Note: I am assuming that Fig 1B and D are in such "early development rescued" larvae rather than true mutants based on the above comment. If so, perhaps a figure like 2A would be useful to clarify this (see minor comment 4 below).
 - How long is transgenic hsp-driven Lh3 robustly expressed in the optic chiasm? Perhaps the authors might demonstrate with a myc stain.
 - Finally - While I agree that it seems most likely that loss of Lh3 is impeding regeneration specifically, could the authors demonstrate or cite that no late-growing axons are crossing the midline at these stages? This would eliminate the possibility that Lh3 mutants have some developmental delay and that presumed regenerating axons are instead late-developing axons that now fail to develop (not regenerate) properly.
- The idea that axons are misrouted in transected Lh3^{-/-} nerves is compelling, but in Figure 3, the caption indicates that this is the case for only half of Lh3 mutant RGC-transected larvae (6/12). What is happening in the other half? Do the nerves stall, indicating a restrictive environment in Lh3 mutants, or do they cross, indicating only a partial requirement for Lh3 to create an instructive environment for regrowth? Parsing the categories would provide more information about how path regrowth is defined by Lh3.
- The data indicating which cells supply Lh3 are intriguing, particularly given that Lh3 appears more strongly expressed in microglia vs. oligodendrocytes, and I am left with a few questions:
 - First, the expression of Lh3 via HCR (Figure 5F) appears much broader than the oligodendrocyte marker. That, plus the likelihood of overexpression via transgenic sox10:Lh3, suggests that endogenous Lh3 might be more broadly expressed to drive axon outgrowth than just from oligodendrocytes. Can the authors use combinatorial labeling with other tissue types to parse what cell types upregulate Lh3 in response to transection?
 - To that end, the data in Figure 5B seem to leave open a possible role for microglia; the control bars (white bars) appear different from those in 5B, perhaps even significantly so (?). There could be a more subtle effect, in which case perhaps additional replicates are necessary.
 - Is there an additive effect? For instance, if Lh3 is supplied by both sox10⁺ and mpeg⁺ glia, is innervation enhanced beyond either single transgenic? Additionally/alternatively, if microglia function is impeded, does that impact axon regeneration even in the context of sox10:Lh3?
- The authors approach a mechanism by demonstrating that the candidate Lh3 substrate Col18a1 is upregulated following transection and that Col18a1 mutants fail to regenerate properly. However, I have a few questions about the interpretations:
 - First, these data alone don't necessarily place them in a pathway, which the authors note in the discussion. It does seem possible, perhaps equally likely that Col18a1 is expressed as part of regenerating the tissue, and that failure to regenerate surrounding tissue (without modification) is

responsible for the phenocopy. The authors would need additional experiments to formally place these in a mechanism; for instance, target presumed Lh3 active sites in col18a1, perform a rescue/heterozygote enhancement assay, etc.

- As noted above, the development/regeneration question is tricky. The authors note possible connections between this gene and eye abnormalities in humans; is there a developmental phenotype in col18a1 mutants? The nerves in Figure 6C do look different from 6B but I can't tell if that is just the angle of the image.

5. One key issue is that data values aren't reported in the text nor figure legends/supplement, and some data reporting is unclear leading to questions about statistical analysis. Specifically:

- For Figures 1-2, I am assuming that the n reported in text is number of nerves (1 per larvae), but it could mean % axons within a nerve, as it's not fully clear.

- The dots represent means (?) from technical replicates, but how many n are measured in each technical replicate?

- The denominator for the percentage isn't clear - is it total transected nerves/larvae? Furthermore, are the categories binary - crossed vs. misguided? Is "stalled" a category?

- On that note, it seems the bars from siblings in panels 1K-L should add up to 100% - same for mutants - but they do not. If these data are paired (e.g. data in K is paired to L) then individual t-tests may not be appropriate.

- Would it be more appropriate to simply report the number of axons crossed/misguided throughout the paper, versus a fraction? If the authors alter their reporting methods earlier in the paper, please be sure to apply to later figures (e.g. 5E, 6J-K).

- In Figure 4, the n of mutants is reported as 5 larvae. How many axons in the graph were from each mutant? Does every mutant have at least one misguided axon? The dots might be color-coded to indicate what's happening at the organismal level and a mixed-methods analysis can be performed between the siblings and mutants.

Minor comments

1. In ZFIN the gene encoding lh3 is called plod3. For consistency, the authors should either follow nomenclature throughout or at minimum name use the alternate gene name at the beginning of the paper for clarity.

2. The authors note that a shelf screen-type study was performed to identify lh3, which was one of the "top hits". However, the other genes are not listed; these would be useful to the field and also to determine by how much lh3 was a "top hit". If the authors are unsure about publishing the results of the screen, perhaps the discussion of the screen should be eliminated in favor of simply noting why lh3 is a good candidate, e.g. discussing more thoroughly its role in peripheral axon regeneration earlier in the paper.

3. While not essential for the message of the study, the authors don't test for functional regeneration, which is feasible and could enhance the study's connection to human eye disorders.

4. The heat shock methods could be more clearly explained or illustrated in the text. For instance, in Fig. 2, do the authors heat shock early in development and later as well? The method of heat shocking is described in methods, but it is hard to map this onto the experiments.

5. There is a typo in Figure 3: "Aberrant" instead of abberant.

6. In Figure 4D, the maximum angle in the y-axis should be -180 to 180 if I understand this measurement correctly.

7. The authors state in the discussion (lines 276) that mpeg+ cells "partially rescue" optic nerve regrowth but this is statistically incorrect (even if I do agree with the sentiment, see above). Formally there is no difference in the mutants with and without mpeg:lh3 expression so there is no partial rescue.

Reviewer 2

Advance summary and potential significance to field

Harvey et al. present a manuscript describing the role of Lh3 and Col18a1 in a larval zebrafish model of optic nerve regeneration. Using the powerful genetic tools available for the zebrafish model, they demonstrate that mutant zebrafish in the lh3 and col18a1 genes have a significant guidance defect during optic nerve regeneration. Timelapse imaging of the regenerating visual system is used to demonstrate the midline growth of wildtype regenerating axons while mutant axons grow into the eye orbit. Both genes are induced after optic nerve injury in putative oligodendrocytes and/or oligodendrocyte precursor cells in the injury and optic chiasm region. Lh3 mutant phenotypes can be rescued with Lh3 overexpression by heat shock inducible transgene. More specifically, they use mpeg1 and sox10 promoter driven transgenes to rescue expression in microglia or OPCs/oligodendrocytes. Only the sox10 promoter driven transgene is sufficient for rescue. Suggesting this lineage is the critical mediator of the lh3/col18a1 effect on regenerating axon guidance to the midline.

Overall, this is a generally well-done study using a highly innovative model system. The observed role of Lh3 and Col18a1 are novel in visual system regeneration independent of any developmental role. Given the difficulty in getting retinal ganglion cells to survive axonal injury and regenerate axons in other models, very little is known about environmental cues in the visual system needed for axon growth and guidance during regeneration. This manuscript directly addresses this gap in knowledge. Mechanistically, the role of Lh3 and collagen in motor axon regeneration has been explored previously by this group. Here they reuse many of the resources/tools generated for those studies to examine the visual system. So mechanistically the observation is less novel. However, the fact that this mechanism is conserved, albeit with different collagen genes, between different CNS and PNS nerves is an important observation. This reviewer has some specific comments to enhance the rigor of the study's conclusions noted below. If these questions can be resolved, I believe this manuscript is a significant advance for the nerve regeneration field.

Comments for the author

1. The statistical analysis is incorrect for much of the data. In previous publications and supplemental figure S1, the authors correctly use the chi square test to compare injury responses of the categorical data. For Fig.1 K,L, Fig.2 F, Fig. 5 B, E, and Fig. 6 J,K the authors use "technical replicates" with n= 3 or 4 although actual sample sizes were in the >10 animal range. It is unclear what this "technical replicate" represents, and the incorrect t-test was applied. I suggest all data is reanalyzed using the chi square test and true biological sample sizes.
2. Similarly, no statistical analysis was presented in Figure 4D. This data could be analyzed for randomness of distribution using the Watson's U-squared test.
3. A major conclusion of the manuscript is that OPC/oligodendrocytes are expressing lh3 and col18a1 with Lh3 likely directly modifying Col18a1 to stimulate axon guidance. However, the resolution of the immunostaining, transgene expression, and HCR in situ hybridization images provided are not high enough to definitively show cellular colocalization. Please add high resolution imaging with 3D reconstructions showing co-localization of lh3 and col18a1 expression in OPC/oligodendrocytes after optic nerve injury. An alternative to demonstrate this may be single cell RNA-seq analysis, although that may be beyond the scope of this study.
4. The conclusion that col18a1 is working with lh3 to support regeneration is more correlational and observational than experimentally demonstrated in the current manuscript. Additional epistasis experiments showing lh3 overexpression is insufficient to rescue col18a1 mutant regeneration and that col18a1 expression is insufficient to rescue lh3 mutant regeneration, while rescuing its own mutant, would significantly strengthen this result. However, this reviewer would accept the co-expression data suggested in point #3 as sufficient evidence that the hypothesis is likely correct.

First revisionAuthor response to reviewers' comments

Comments from the Reviewers:

Reviewer 1: SUMMARY OF THE ADVANCE MADE IN THIS PAPER AND ITS POTENTIAL SIGNIFICANCE TO THE FIELD

In this manuscript, Harvey et al. identify the glycosyltransferase Lh3 as essential for directional outgrowth of retinal ganglion cell (RGC) axons following optic nerve injury. The authors demonstrate that lh3 is upregulated by neighboring tissues following nerve transection and that transgenically supplying lh3 from oligodendrocyte lineage cells is sufficient for restoration of directional axon guidance. Finally, the authors suggest that col18a1, which is also upregulated following transection and essential for proper nerve regeneration, is subject to modification by Lh3 to promote optic nerve regeneration. Altogether, this work identifies a pro-regenerative pathway in the CNS that is necessary to steer RGC axons along the proper path following nerve transection. This work also identifies comparable mechanisms between PNS and CNS environments, as previous work from this group found that Lh3 is also necessary for regenerating motor axons via peripheral glia-mediated pathways. Altogether, this work builds on previous studies to demonstrate recapitulation of Lh3 function across tissues to promote axon regeneration.

SUGGESTIONS TO AUTHORS

Major comments

1. The authors discuss bypassing the requirement for lh3 during development by using a validated heat-shock promoter driving Lh3 beginning early in development, as described in methods. This note suggests that perhaps there is a developmental requirement for Lh3 by RGC axons that is not yet described, at least not in the cited papers. I have concerns that this might confound the take-home messages of this study.

- What is the optic nerve phenotype of lh3 mutants without hsp:lh3 expression? Note: I am assuming that Fig 1B and D are in such "early development rescued" larvae rather than true mutants based on the above comment. If so, perhaps a figure like 2A would be useful to clarify this (see minor comment 4 below).

Without hsp:lh3 induction, the RGC axon projections in *lh3* mutants are indistinguishable from siblings. This has been previously described and shown in Figure 4 of Zeller et al., 2002, doi: 10.1006/dbio.2002.0852. The reviewer is correct that the mutants shown in Figure 1 receive an early developmental induction, and so we have added a timeline to Figure 1 to help clarify the experimental timelines. We also added text to Lines 99-105 to clarify that the developmental heat shocks are performed to rescue other morphological defects in *lh3* mutant embryos, rather than to rescue any deficits in the retinotectal projection.

- How long is transgenic hsp-driven Lh3 robustly expressed in the optic chiasm? Perhaps the authors might demonstrate with a myc stain.

We previously performed western blots (Figure S1., Isaacman-Beck et al., Neuron, 2015) and showed that Lh3 is no longer detectable via western blot by 4 days post fertilization following early developmental induction of Lh3 expression from the transgenic line Tg(hsp70:lh3-myc). It is therefore unlikely that significant levels of Lh3 expression persist to 5 dpf when we perform our transections. Thus our experiments in this submission show that Lh3 functions during the time period of RGC axon regeneration.

- Finally - While I agree that it seems most likely that loss of lh3 is impeding regeneration specifically, could the authors demonstrate or cite that no late-growing axons are crossing the midline at these stages? This would eliminate the possibility that lh3 mutants have some developmental delay and that presumed regenerating axons are instead late-developing axons that now fail to develop (not regenerate) properly.

We cannot exclude the possibility that during the course of our experiments a small number of newly born RGCs from retinal stem or progenitor cells emerge. It is well established that zebrafish retinas continue to grow through the lifespan of the fish (Marcus et al., 1999 doi: 10.1017/s095252389916303x.) However, we have previously shown in Harvey et al., Plos One, 2019 that our optic nerve transection assay does not lead to detectable levels of RGC proliferation. Furthermore, in preliminary experiments, we have retrogradely labeled axons initiating regrowth in

WT larvae (not shown here) and find that those RGCs are not in or near the ciliary marginal zone, the location of retinal stem cell niche. This strongly suggests that regenerating axons, in particular those growing during the initial stages of regeneration, are not a late developing RGC population.

2. The idea that axons are misrouted in transected *lh3*^{-/-} nerves is compelling, but in Figure 3, the caption indicates that this is the case for only half of *lh3* mutant RGC-transected larvae (6/12). What is happening in the other half? Do the nerves stall, indicating a restrictive environment in *lh3* mutants, or do they cross, indicating only a partial requirement for *lh3* to create an instructive environment for regrowth? Parsing the categories would provide more information about how path regrowth is defined by *Lh3*.

In response to this comment, as well as subsequent comments from Review 1 and 2, we have displayed our quantification of regenerating nerves as stacked bar graphs to display three categories in which transected nerves are scored at 72hpt: Tectal Innervation Across the Midline, Aberrant Growth and Stalled Growth. We have included a graph that shows the categories of all of the nerves that were timelapsed in a new Figure S1.

3. The data indicating which cells supply *Lh3* are intriguing, particularly given that *Lh3* appears more strongly expressed in microglia vs. oligodendrocytes, and I am left with a few questions:

- First, the expression of *lh3* via HCR (Figure 5F) appears much broader than the oligodendrocyte marker. That, plus the likelihood of overexpression via transgenic *sox10:lh3*, suggests that endogenous *lh3* might be more broadly expressed to drive axon outgrowth than just from oligodendrocytes. Can the authors use combinatorial labeling with other tissue types to parse what cell types upregulate *lh3* in response to transection?

We agree with Reviewer 1's observation regarding Figure 5F that the expression pattern of *lh3* mRNA extends beyond that of *Tg(olig2:dsred)*. This suggests that *Lh3* might be acting in additional cell types present within the optic chiasm. However, identifying these cells or their precise function is outside the scope of this manuscript.

- To that end, the data in Figure 5B seem to leave open a possible role for microglia; the control bars (white bars) appear different from those in 5B, perhaps even significantly so (?). There could be a more subtle effect, in which case perhaps additional replicates are necessary.

We are excited to report that we performed additional replicates of the experiment in (what was) Figure 5B. The results more definitively demonstrate that expressing *Lh3* in microglia/macrophages fails to rescue the *lh3* mutant phenotype. We hope this is satisfactory for Reviewer 1.

- Is there an additive effect? For instance, if *lh3* is supplied by both *sox10*⁺ and *mpeg*⁺ glia, is innervation enhanced beyond either single transgenic? Additionally/alternatively, if microglia function is impeded, does that impact axon regeneration even in the context of *sox10:lh3*?

We acknowledge that we cannot completely exclude a role for *Lh3* in microglia/macrophages together with oligodendrocytes in optic nerve regeneration. While we find a potential role of microglia in this regeneration paradigm very intriguing, we believe that combining the transgenes would not enhance the observed rescue since expression of *Lh3* driven by *sox10* almost completely rescues the *lh3* mutant phenotype to sibling levels of regeneration (Figure 5E). For the purpose of this study, we use the transgenic expression of *Lh3* in macrophages to show that expressing *Lh3* in any cell type that is present near regenerating RGC axons is insufficient to rescue the *lh3* mutant phenotype.

4. The authors approach a mechanism by demonstrating that the candidate *Lh3* substrate *Col18a1* is upregulated following transection and that *col18a1* mutants fail to regenerate properly. However, I have a few questions about the interpretations:

- First, these data alone don't necessarily place them in a pathway, which the authors note in the discussion. It does seem possible, perhaps equally likely that *col18a1* is expressed as part of regenerating the tissue, and that failure to regenerate surrounding tissue (without modification) is responsible for the phenocopy. The authors would need additional experiments to formally place these in a mechanism; for instance, target presumed *Lh3* active sites in *col18a1*, perform a rescue/heterozygote enhancement assay, etc.

We agree that additional experiments would be needed to demonstrate that *Lh3* acts through *Col18a1* to direct Line 320 "...we find in optic nerve regeneration that *Lh3* potentially acts through *Col18a1*...". Such experiments would require use to be able to monitor *Lh3* mediated

Col18a1 posttranslational modification. Generating the tools and assays required for these experiments, while very informative, will require extended periods of time, which is outside the timeline for this manuscript.

- As noted above, the development/regeneration question is tricky. The authors note possible connections between this gene and eye abnormalities in humans; is there a developmental phenotype in *col18a1* mutants? The nerves in Figure 6C do look different from 6B but I can't tell if that is just the angle of the image.

We find that after immunostaining and mounting larvae between slides and coverslips, RGC axons at the chiasm sometimes resemble a slanted X, rather than a distinct X. We attribute this variability to mounting rather than reflecting a phenotype. This variability from mounting the samples can also be seen when comparing images in Figure 1. Most importantly, similar to *lh3* mutants, prior to injury, RGC axon projections in *col18a1* mutants are indistinguishable from those in siblings, consistent with the idea that both genes are required for RGC axon regeneration rather than RGC development.

5. One key issue is that data values aren't reported in the text nor figure legends/supplement, and some data reporting is unclear leading to questions about statistical analysis. Specifically:

- For Figures 1-2, I am assuming that the n reported in text is number of nerves (1 per larvae), but it could mean % axons within a nerve, as it's not fully clear.

- The dots represent means (?) from technical replicates, but how many n are measured in each technical replicate?

- The denominator for the percentage isn't clear - is it total transected nerves/larvae? Furthermore, are the categories binary - crossed vs. misguided? Is "stalled" a category?

- On that note, it seems the bars from siblings in panels 1K-L should add up to 100% - same for mutants - but they do not. If these data are paired (e.g. data in K is paired to L) then individual t-tests may not be appropriate.

- Would it be more appropriate to simply report the number of axons crossed/misguided throughout the paper, versus a fraction? If the authors alter their reporting methods earlier in the paper, please be sure to apply to later figures (e.g. 5E, 6J-K).

We thank the Reviewer for pointing out the lack of clarity regarding the phenotypic quantification. To clarify our quantifications, we have displayed our quantification of regenerating nerves as stacked bar graphs to display three categories in which transected nerves are scored at 72hpt: Tectal Innervation Across the Midline, Aberrant Growth as well as Stalled Growth. We transect 2 nerves per larvae (2 eyes) and then score how each nerve regenerates. So, the numbers displayed on each bar in a graph are the total number of nerves comprising that bar, not axons. For Figure 1, and subsequent figures, all nerves are included in the same graph, whereas in our initial submission, we did not include Stalled Nerves in any graphs.

- In Figure 4, the n of mutants is reported as 5 larvae. How many axons in the graph were from each mutant? Does every mutant have at least one misguided axon? The dots might be color-coded to indicate what's happening at the organismal level and a mixed-methods analysis can be performed between the siblings and mutants.

In order to address this comment, we included Table 1, which contains the raw angles measured for each fascicle used in Figure 4D. Importantly, single optic nerves consist of hundreds of RGC axons that are capable of tightly fasciculating with each other. We use bulk labeling of the RGC axons using the *Tg(isl2b:GFP)*, which precludes single axon analysis and is why we use the term "fascicles".

Minor comments

1. In ZFIN the gene encoding *lh3* is called *plod3*. For consistency, the authors should either follow nomenclature throughout or at minimum name use the alternate gene name at the beginning of the paper for clarity.

We refer to the *plod3* gene in the abstract and introduction for clarity.

2. The authors note that a shelf screen-type study was performed to identify *lh3*, which was one of the "top hits". However, the other genes are not listed; these would be useful to the field and also to determine by how much *lh3* was a "top hit". If the authors are unsure about publishing the results of the screen, perhaps the discussion of the screen should be eliminated in favor of simply noting

why *lh3* is a good candidate, e.g. discussing more thoroughly its role in peripheral axon regeneration earlier in the paper.

The reviewer raises an important point. As the reviewer points out, this manuscript is not about a screen. Instead, we selected two mutants from this screen and characterized their roles at the molecular and cellular level. We believe that without characterizing all other 'hits' from the screen in more detail, it is impossible to rank the 'hits' against each other. We agree that the other genes would be helpful to the field, and we are actively working on further characterizing them. We believe that it would be premature to even list them in this manuscript as this would trigger many questions that we are currently unable to answer.

3. While not essential for the message of the study, the authors don't test for functional regeneration, which is feasible and could enhance the study's connection to human eye disorders.

We have previously shown that functional regeneration in this optic nerve regeneration assay occurs by 8 days post transection in Harvey et al., 2019. In this manuscript, we focused on the early stages of the regeneration. While an interesting question, performing functional regeneration assays seems outside the scope of the goal of this study.

4. The heat shock methods could be more clearly explained or illustrated in the text. For instance, in Fig. 2, do the authors heat shock early in development and later as well? The method of heat shocking is described in methods, but it is hard to map this onto the experiments.

We have updated the timeline in Figure 2 to help clarify the strategy we employed.

5. There is a typo in Figure 3: "Aberrant" instead of abberant.

Thank you, we have made the edit.

6. In Figure 4D, the maximum angle in the y-axis should be -180 to 180 if I understand this measurement correctly.

We thank the reviewer for raising this point. To most effectively display the data and simplify the graph, we now display the absolute value of the angles from the original path. We also include additional categories of nerves from *lh3* siblings and mutants for completion. We also include the raw angle values in Table 1.

7. The authors state in the discussion (lines 276) that *mpeg+* cells "partially rescue" optic nerve regrowth but this is statistically incorrect (even if I do agree with the sentiment, see above). Formally there is no difference in the mutants with and without *mpeg:lh3* expression so there is no partial rescue.

We agree with the Reviewer's correction, and have adjusted the text accordingly.

Reviewer 2: SUMMARY OF THE ADVANCE MADE IN THIS PAPER AND ITS POTENTIAL SIGNIFICANCE TO THE FIELD

Harvey et al. present a manuscript describing the role of *Lh3* and *Col18a1* in a larval zebrafish model of optic nerve regeneration. Using the powerful genetic tools available for the zebrafish model, they demonstrate that mutant zebrafish in the *lh3* and *col18a1* genes have a significant guidance defect during optic nerve regeneration. Timelapse imaging of the regenerating visual system is used to demonstrate the midline growth of wildtype regenerating axons while mutant axons grow into the eye orbit. Both genes are induced after optic nerve injury in putative oligodendrocytes and/or oligodendrocyte precursor cells in the injury and optic chiasm region. *Lh3* mutant phenotypes can be rescued with *Lh3* overexpression by heat shock inducible transgene. More specifically, they use *mpeg1* and *sox10* promoter driven transgenes to rescue expression in microglia or OPCs/oligodendrocytes. Only the *sox10* promoter driven transgene is sufficient for rescue. Suggesting this lineage is the critical mediator of the *lh3/col18a1* effect on regenerating axon guidance to the midline.

Overall, this is a generally well-done study using a highly innovative model system. The observed role of *Lh3* and *Col18a1* are novel in visual system regeneration independent of any developmental role. Given the difficulty in getting retinal ganglion cells to survive axonal injury and regenerate axons in other models, very little is known about environmental cues in the visual system needed for axon growth and guidance during regeneration. This manuscript directly addresses this gap in

knowledge. Mechanistically, the role of Lh3 and collagen in motor axon regeneration has been explored previously by this group. Here they reuse many of the resources/tools generated for those studies to examine the visual system. So mechanistically the observation is less novel. However, the fact that this mechanism is conserved, albeit with different collagen genes, between different CNS and PNS nerves is an important observation. This reviewer has some specific comments to enhance the rigor of the study's conclusions noted below. If these questions can be resolved, I believe this manuscript is a significant advance for the nerve regeneration field.

SUGGESTIONS TO AUTHORS

1. The statistical analysis is incorrect for much of the data. In previous publications and supplemental figure S1, the authors correctly use the chi square test to compare injury responses of the categorical data. For Fig.1 K,L, Fig.2 F, Fig. 5 B, E, and Fig. 6 J,K the authors use "technical replicates" with n= 3 or 4 although actual sample sizes were in the >10 animal range. It is unclear what this "technical replicate" represents, and the incorrect t-test was applied. I suggest all data is reanalyzed using the chi square test and true biological sample sizes.

We have followed the suggestion made and changed the quantification of regenerating nerves to stacked bar graphs to display three categories in which transected nerves are scored at 72hpt: Tectal Innervation Across the Midline, Aberrant Growth as well as Stalled Growth. This is similar to how Figure S1 was displayed in our original submission. Chi square tests are performed to determine statistical significance.

2. Similarly, no statistical analysis was presented in Figure 4D. This data could be analyzed for randomness of distribution using the Watson's U-squared test.

Prompted by the reviewers' comments, we have reassessed how to most effectively display this data. We have changed the angles to absolute angles from the original path to simplify the graph. We also include additional categories of nerves from *lh3* siblings and mutants for completion.

3. A major conclusion of the manuscript is that OPC/oligodendrocytes are expressing *lh3* and *col18a1* with *Lh3* likely directly modifying *Col18a1* to stimulate axon guidance. However, the resolution of the immunostaining, transgene expression, and HCR in situ hybridization images provided are not high enough to definitively show cellular colocalization. Please add high resolution imaging with 3D reconstructions showing co-localization of *lh3* and *col18a1* expression in OPC/oligodendrocytes after optic nerve injury. An alternative to demonstrate this may be single cell RNA-seq analysis, although that may be beyond the scope of this study.

We have performed additional in situ hybridizations and imaged on a Zeiss 880 using Airyscan superresolution for new Figures 5G and 6B. We also include 3D rotations of these stacks as Supplemental Videos 3 and 4.

4. The conclusion that *col18a1* is working with *lh3* to support regeneration is more correlational and observational than experimentally demonstrated in the current manuscript. Additional epistasis experiments showing *lh3* overexpression is insufficient to rescue *col18a1* mutant regeneration and that *col18a1* expression is insufficient to rescue *lh3* mutant regeneration, while rescuing its own mutant, would significantly strengthen this result. However, this reviewer would accept the co-expression data suggested in point #3 as sufficient evidence that the hypothesis is likely correct.

We look forward to performing the additional rescue experiments in future studies to further demonstrate that *Col18a1* is a direct substrate of *Lh3*, as Reviewer's 2 describes. For this study, we appreciate Reviewer's 2 comment and have included the co-expression data as mentioned above.

Second decision letter

MS ID#: dev.205048R1

MS TITLE: A glial cell derived pathway directs regenerating optic nerve axons toward the optic chiasm

AUTHORS: Beth Harvey, Melissa Baxter, Alexis M. Garcia and Michael Granato

Dear Beth and Michael,

I have now received all the referees reports on the above manuscript. The referees' comments are appended below.

The overall evaluation is positive and the reviewers just have some relatively minor points to address prior to publication. Please attend to all of the reviewers' comments in your revised manuscript and detail them in your point-by-point response.

Reviewer 1

Advance summary and potential significance to field

As described in the previous review, Harvey et al. identify the glycosyltransferase Lh3 as essential for directional outgrowth of retinal ganglion cell (RGC) axons following optic nerve injury. The authors demonstrate that Lh3 is upregulated by neighboring tissues following nerve transection, and transgenically supplying Lh3 from oligodendrocyte lineage cells is sufficient for restoration of directional axon guidance. Finally, the authors suggest that COL18A1, which is also upregulated following transection and essential for proper nerve regeneration, is subject to modification by Lh3 to promote optic nerve regeneration. Altogether, this work identifies and characterizes an important pro-regenerative pathway in the CNS.

This revised version strengthens the "glial cell derived" aspect of the work, though the authors might still comment on comparable mechanisms of Lh3 in PNS and CNS environments. Data reporting and visualization are much improved in this version, and the statistical analyses now appear acceptable for these data. I think my suggestions at this point are sufficiently minor and text-based so they might be reviewed by the editor alone.

Comments for the author

Scientific comments:

1. It is understandable that the authors do not want to describe the results of their candidate screen. But they describe a "candidate screen" without really explaining why Lh3 was a candidate - frankly, they wouldn't even need to mention the screen, as Lh3 is a good candidate by itself. The authors note they screened candidates either expressed in visual system OR with roles in axon guidance (not both?) therefore obscuring the role of Lh3 in PNS regeneration until later in the paper. So while they don't need to describe the screen, the authors should at least provide context for Lh3 "candidacy" earlier. This will not undermine the novelty as it is important to identify factors that can promote regeneration both in optic nerve (part of CNS) and peripheral nerves, and oligodendrocytes are sufficiently different from Schwann cells (particularly during regeneration) that this is an intriguing point that enhances the value of the paper.
2. The authors are appropriately circumspect with their claims about COL18A1 as a substrate for Lh3 in this process. However, the broad expression they identified is relevant for an alternative (or additional) model. I think it is ok for the authors to save this investigation for a future study, but they should at least note around line 308 that this model is consistent with the broad expression of COL18A1 they observed in Figure 6B.

Additional comments that would enhance the quality and readability of the manuscript:

1. Verb tense is sometimes inconsistent in the Results section, e.g. in lines 118-118, axons that "did or did not receive... regrow... and innervated" mixes past and present tense.
2. Figure panels might be listed (lettered) in the order they appear in the text; e.g. Fig 5D is introduced in line 188 while 5B is mentioned in 191.

Reviewer 2

Advance summary and potential significance to field

The authors have addressed the majority of this reviewer's concerns with clarifications and additional experiments. The manuscript represents a significant advance in our understanding of the molecular/genetic mechanisms of optic nerve regeneration. I have two minor clarifications noted below.

Comments for the author

Figure 2F - Why is no statistical comparison made between "No Regeneration Heat Shock (-/-)" and "With Regeneration Heat Shock (-/-)" groups? This is the definitive rescue comparison. A similar comparison was done in Figure 5 D and E for a different experiment. Please add this information and comment on its statistical significance or lack thereof.

Figure 4D - The authors' suggest the data in the "aberrant growth group" demonstrates a "greater range of angles in the mutant versus the wt/het. This certainly appears to be the case but a statistical comparison should be made.

Second revision

Author response to reviewers' comments

Comments from the Reviewers:

Reviewer 1: SUMMARY OF THE ADVANCE MADE IN THIS PAPER AND ITS POTENTIAL SIGNIFICANCE TO THE FIELD

As described in the previous review, Harvey et al. identify the glycosyltransferase Lh3 as essential for directional outgrowth of retinal ganglion cell (RGC) axons following optic nerve injury. The authors demonstrate that lh3 is upregulated by neighboring tissues following nerve transection, and transgenically supplying lh3 from oligodendrocyte lineage cells is sufficient for restoration of directional axon guidance. Finally, the authors suggest that col18a1, which is also upregulated following transection and essential for proper nerve regeneration, is subject to modification by Lh3 to promote optic nerve regeneration. Altogether, this work identifies and characterizes an important pro-regenerative pathway in the CNS.

This revised version strengthens the "glial cell derived" aspect of the work, though the authors might still comment on comparable mechanisms of lh3 in PNS and CNS environments. Data reporting and visualization are much improved in this version, and the statistical analyses now appear acceptable for these data. I think my suggestions at this point are sufficiently minor and text-based so they might be reviewed by the editor alone.

SUGGESTIONS TO AUTHORS

Scientific comments:

1. It is understandable that the authors do not want to describe the results of their candidate screen. But they describe a "candidate screen" without really explaining why lh3 was a candidate - frankly, they wouldn't even need to mention the screen, as lh3 is a good candidate by itself. The authors note they screened candidates either expressed in visual system OR with roles in axon guidance (not both?) therefore obscuring the role of lh3 in PNS regeneration until later in the paper. So while they don't need to describe the screen, the authors should at least provide context for lh3 "candidacy" earlier. This will not undermine the novelty as it is important to identify factors that can promote regeneration both in optic nerve (part of CNS) and peripheral nerves, and oligodendrocytes are sufficiently different from Schwann cells (particularly during regeneration) that this is an intriguing point that enhances the value of the paper.

After further consideration, we agree that adding additional context for choosing Lh3 as a candidate earlier in the manuscript helps to clarify our rationale. To address these comments made by the Reviewer, we have added text at Lines 99-104 saying "One gene of particular interest to us was lh3, which encodes a multidomain glycosyltransferase that post-translationally modifies collagen proteins (Heikkinen et al., 2000; C. Wang et al., 2002). We have previously shown that Lh3 is required in Schwann cells to promote target-specific regeneration of dorsal motor nerve axons in

the peripheral nervous system (Isaacman-Beck et al., 2015). We therefore were interested in examining a potential similar role for Lh3 in optic nerve regeneration.”

2. The authors are appropriately circumspect with their claims about col18a1 as a substrate for lh3 in this process. However, the broad expression they identified is relevant for an alternative (or additional) model. I think it is ok for the authors to save this investigation for a future study, but they should at least note around line 308 that this model is consistent with the broad expression of col18a1 they observed in Figure 6B.

To address these comments made by the Reviewer, we have added text at now Line 313 saying “Using in situ hybridizations, we observed *col18a1* expression surrounding the injured axon stumps but also expressed somewhat broadly in the chiasm region (Fig 6B). We therefore cannot exclude a possible mechanism involving RGC axons directly interacting with Col18a1 in the extracellular matrix through integrins (Elango et al., 2022).”

Additional comments that would enhance the quality and readability of the manuscript:

1. Verb tense is sometimes inconsistent in the Results section, e.g. in lines 118-118, axons that “did or did not receive... regrow... and innervated” mixes past and present tense.

We thank the reviewer for bringing this to our attention, and have gone through the entire manuscript to edit the verb tense for consistency when referring to our results in this manuscript versus conclusive statements. Please see all of the highlighted regions in the manuscript.

2. Figure panels might be listed (lettered) in the order they appear in the text; e.g. Fig 5D is introduced in line 188 while 5B is mentioned in 191.

To address this comment, we have slightly adjusted the panels and edited the panel labels in Figure 5 to change the order in which they appear in the text. To also improve other figures and remain consistent, we have changed the panel labels (only) in both Figure 1 and 6. We went back into the text and Figure legends to make sure everything was appropriately changed to match.

Reviewer 2: SUMMARY OF THE ADVANCE MADE IN THIS PAPER AND ITS POTENTIAL SIGNIFICANCE TO THE FIELD

The authors have addressed the majority of this reviewer's concerns with clarifications and additional experiments. The manuscript represents a significant advance in our understanding of the molecular/genetic mechanisms of optic nerve regeneration. I have two minor clarifications noted below.

SUGGESTIONS TO AUTHORS

Figure 2F - Why is no statistical comparison made between “No Regeneration Heat Shock (-/-)” and “With Regeneration Heat Shock (-/-)” groups? This is the definitive rescue comparison. A similar comparison was done in Figure 5 D and E for a different experiment. Please add this information and comment on its statistical significance or lack thereof.

We have included the additional statistical analysis between the “No Regeneration Heat Shock (-/-)” and “With Regeneration Heat Shock (-/-)” groups in Figure 2F. We also want to point out that we repeated the statistical analysis between the siblings and mutants with the Regeneration Heat Shock, and corrected the significance from ** to *, $p=0.0279$, Fisher's exact test in the Figure legend at Line 759.

Figure 4D - The authors' suggest the data in the “aberrant growth group” demonstrates a “greater range of angles in the mutant versus the wt/het. This certainly appears to be the case but a statistical comparison should be made.

We appreciate the reviewer's suggestion, and have performed an additional statistical test. We included this text at Line 174 “comparing lh3 siblings ‘Tectal Innervation’ to lh3 mutants ‘Aberrant Growth,’ $p=0.0005$, f-test to compare variances”

Third decision letter

MS ID#: dev.205048R2

MS TITLE: A glial cell derived pathway directs regenerating optic nerve axons toward the optic chiasm

AUTHORS: Beth Harvey, Melissa Baxter, Alexis M. Garcia and Michael Granato

Dear Beth and Michael,

I am happy to tell you that your manuscript has been accepted for publication in Development, pending our standard publication integrity checks.